# The impact of diabetes mellitus medication on the incidence of endogenous endophthalmitis

Ke-Hung Chien[1,2], Ke-Hao Huang[1,3], Chi-Hsiang Chung[4,5], Yun-Hsiu Hsieh[1], Chang-Min Liang[1], Yu-Hua Chang[6], Tzu-Heng Weng[1]*, Wu-Chien Chien[4,7,8]*

1 Department of Ophthalmology, Tri-Service General Hospital, National Defense Medical Center, Taipei, Taiwan, 2 Institute of Pharmacology, National Yang-Ming University, Taipei, Taiwan, 3 Department of Ophthalmology, Songshan Branch of Tri-Service General Hospital, National Defense Medical Center, Taipei, Taiwan, 4 School of Public Health, National Defense Medical Center, Taipei, Taiwan, 5 Taiwanese Injury Prevention and Safety Promotion Association (TIPSPA), Taipei, Taiwan, 6 Division of Colorectal Surgery, Department of Surgery, Tri-Service General Hospital, National Defense Medical Center, Taipei, Taiwan, 7 Department of Medical Research, Tri-Service General Hospital, National Defense Medical Center, Taipei, Taiwan, 8 Graduate Institute of Life Science, National Defense Medical Center, Taipei, Taiwan

* aheng7435@gmail.com (THW); chienwu@ndmctsgh.edu.tw (WCC)

## Abstract

### Purpose

This study aimed to evaluate the relationship between diabetic mellitus (DM) treatment and the incidence rate of endogenous endophthalmitis (EE).

### Design

This study used a matched cohort design. We utilized the Longitudinal Health Insurance Database to identify outpatients and inpatients who were diagnosed with DM and treated with medication from 2000 to 2010.

### Methods

Several factors and different DM medications were also investigated. The influence of DM medication on the incidence of EE was examined by using Cox proportional hazards regression models, and the hazard ratios and 95% confidence intervals were determined.

### Results

The cumulative incidence of EE was lower in DM patients treated with medication than in subjects in the control group (P = 0.002). The adjusted hazard ratio (AHR) was 0.47-fold lower in the treatment group than in the control group (P = 0.004). With respect to DM medication, single-agent therapy with insulin, metformin, gliclazide, glimepiride, or repaglinide and combination therapy with glimepiride/metformin or repaglinide/metformin were associated with decreased AHRs (0.257–0.544, all P<0.05).

### Conclusions

Diabetic patients treated with medication had lower AHRs than those in the control group. Further stratification indicated that liver abscess, liver disease DM patients who were treated

**Data Availability Statement:** All relevant data are within the manuscript and its Supporting Information files.

**Funding:** This study was supported by grants from the Tri-Service General Hospital Research Foundation (TSGHC107-004 and TSGH-C107-090) and the Ministry of National Defense Medical Affairs Bureau (MAB-108-049). The sponsors had no roles in the study design, data collection and analysis, decision to publish, or preparation of the manuscript.

**Competing interests:** The authors have declared that no competing interests exist.

**Abbreviations:** DM, diabetes mellitus; EE, endogenous endophthalmitis; NHIRD, National Health Insurance Research Database; CCI, Charlson Comorbidity Index; ICD-9-CM, International Classification of Disease, 9th Revision, Clinical Modification; NT$, new Taiwan dollars; AHR, adjusted hazard ratio.

with medication had a lower risk of developing EE. Several specific DM medications may decrease the incidence of EE.

## Introduction

Endophthalmitis is an inflammatory intraocular condition that involves both the posterior and anterior segments. In most cases, it is caused by an intraocular infection and often results in severe and irreversible visual deterioration [1–3]. Endophthalmitis can be further divided into endogenous and exogenous endophthalmitis based on its pathogenesis. The former is caused by the hematogenous dissemination of bacterial, fungal, or other pathogens that break through the blood-ocular barrier and inoculate the intraocular region [4–7]. In addition, patients with a compromised immune system are at risk of endogenous endophthalmitis (EE). Some of these predisposing conditions include diabetes mellitus (DM), systemic malignancy, endocarditis, sickle cell anemia, autoimmune disease, and human immunodeficiency virus infection/acquired immunodeficiency syndrome (HIV/AIDS) [5,7–9]. Among these diseases, DM is the most often mentioned and discussed. It has been established that impaired neutrophilic bactericidal function is strongly associated with poor glycemic control [10,11]. Some studies have also demonstrated that DM alters the corneal epithelial basement membrane resulting in basal cell degeneration that manifests clinically as superficial punctate keratitis, which causes greater fragility of the ocular barrier [12,13].

In our previous study, several predictors of mortality among inpatients with endophthalmitis were found. Among them, DM resulted in a decreased odds ratio for inpatient mortality [14]. However, studies that have aimed to explore the relationship between DM treatment and the incidence rate of EE are extremely rare. In the present study, we not only analyzed this issue but also stratified several factors and different medication combinations used to treat DM patients by utilizing the National Health Insurance Research Database (NHIRD) in Taiwan over an 11-year span.

## Materials and methods

### Research database

The National Health Insurance (NHI) program in Taiwan was established in March 1995 to include more than 99% of the residents of Taiwan (approximately 23 million people). The NHIRD is derived from the claims data of the NHI program, is available to the public in electronic format for research purposes, and includes all forms of inpatient, outpatient, and emergency health care services.

### Study participants

In the present study, we used the NHIRD to identify outpatients and inpatients in the Longitudinal Health Insurance Database (LHID) who were newly diagnosed with DM according to the International Classification of Disease, 9th Revision, Clinical Modification (ICD-9-CM) code (250 for DM; 360.00–02 for endophthalmitis; and 572 for liver abscess) and were treated with medication from 2000 to 2010. As shown in Fig 1, 26,085 individuals were initially identified, and 1,725 patients were excluded because they had been taking DM medication before 2000, had endophthalmitis diagnosed before 2000, were younger than 18 years old and/or had an unknown sex. A comparison group that was four times larger than the treatment group

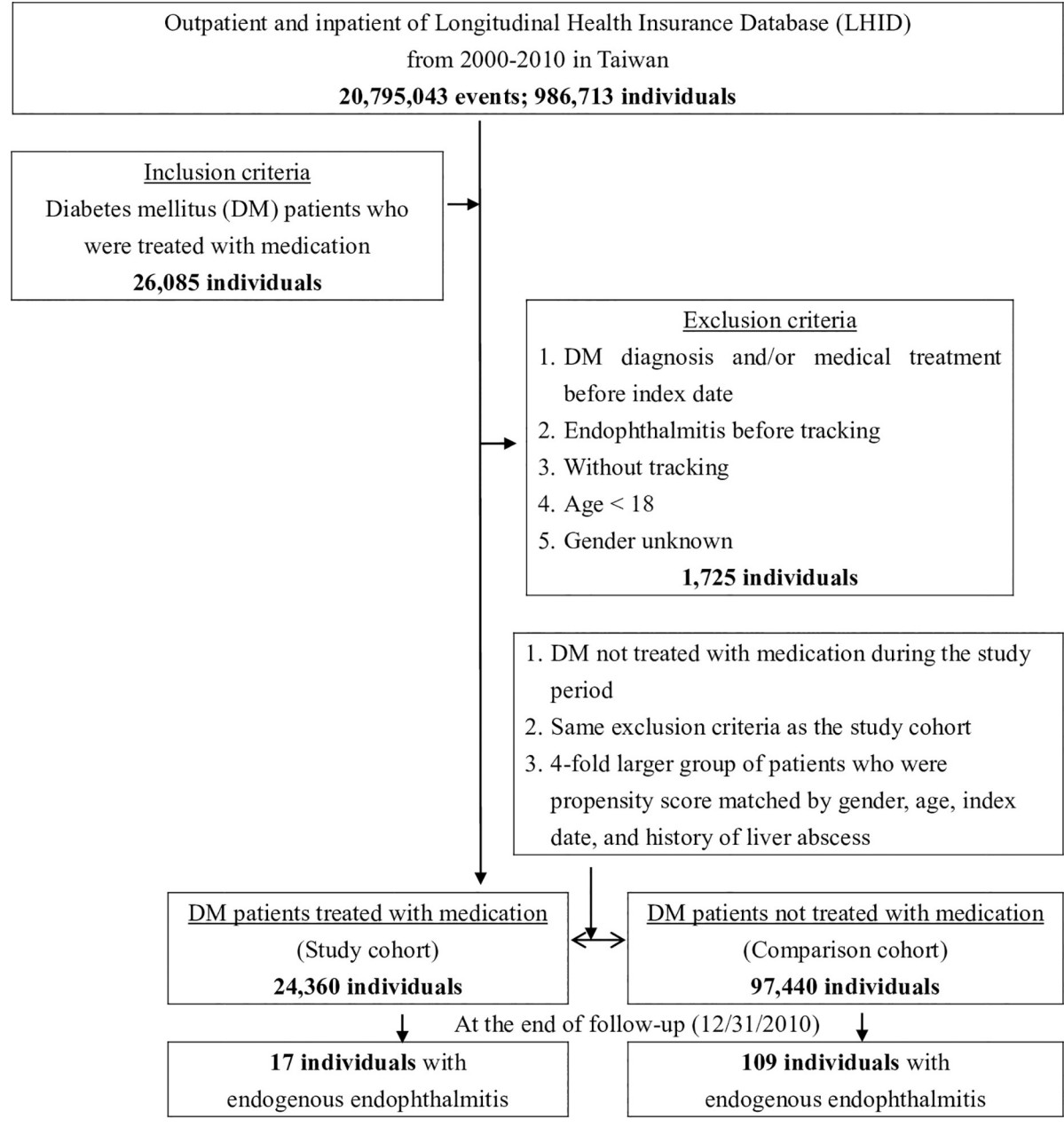

**Fig 1. Flowchart of study sample selection from the National Health Insurance Research Database in Taiwan.**

included individuals who were propensity score matched by sex, age, index date, and history of liver abscesses. To differentiate EE from exogenous endophthalmitis, patients who received intraocular surgeries 6 weeks before the diagnosis of EE were eliminated. The study protocol conformed to the ethical guidelines of the 1975 Declaration of Helsinki.

## Ethical considerations

The NHIRD provides encrypted personal information. Researchers are required to sign a written agreement declaring that they will not violate patient privacy. Patient consent is not

required to access the NHIRD data. The Institutional Review Board of the Tri-Service General Hospital approved this study and waived the consent requirement (TSGHIRB No. 2-105-05-082).

## Statistical analysis

Characteristics of the study population at baseline and at the end of follow-up were analyzed. Several factors and different medical DM therapies were further stratified and investigated. We used the mean±standard deviation (SD) to express continuous variables, and normally distributed continuous data related to DM medications or not were compared between groups using Student's *t*-test. Categorical variables were compared using either Pearson's chi-square test or Fisher's exact test. The latter was used in rare cases in which the expected result was less than 5%. The influence of DM medication on the incidence of EE was examined by using Cox proportional hazards regression models, with hazard ratios and 95% confidence intervals. Statistical significance was defined as P<0.05. All statistical analyses were performed using SPSS software, Version 22.0.

## Results

From 2000 to 2010, we identified 26,085 patients who were diagnosed with and treated for DM out of a total of 986,713 outpatients and hospitalized patients (20,795,043 events) from the LHID in Taiwan. We excluded 1,725 patients because they were diagnosed with DM before 2000, were diagnosed with endophthalmitis before tracking, were under 18 years old, or were of unknown sex. Of the remaining 24,360 patients, EE occurred in 17 individuals over the span of 11 years of tracking (from Jan. 1, 2000, to Dec. 31, 2010). The incidence rate was 0.07% (17/24,360). We also identified a comparison group that was four times greater than the control group (97,440 individuals); this group included individuals who were newly diagnosed with DM and were not treated with medication, using the same exclusion criteria as mentioned above, and who were matched by sex, age, index date and history of liver abscesses. In this comparison group that was followed-up for 11 years, 109 individuals (incidence rate, 0.11%) were subsequently diagnosed with EE (Fig 1).

The Kaplan-Meier (KM) plot in Fig 2 shows the cumulative incidence of EE stratified by medication use, demonstrating that DM patients who were treated with medication had a lower risk of developing EE (log-rank test, P = 0.002).

Of the total 121,800 patients with DM, 24,360 were treated with medication (20%) and 97,440 were not (80%, 4-fold greater number of patients than in the treatment group). The groups were sex- and age-matched (P = 0.999) at the baseline of the study. The comorbidity of liver abscessation was also controlled for, and there was no significant difference between these 2 groups at baseline. However, comorbidities of hypertension, depression, renal disease, and tumors occurred less frequently in patients with DM who were treated with medication than in those the control group at the baseline of the study. Comorbidities of hyperlipidemia and thyrotoxicosis occurred more frequently in patients with DM who were treated with medication than in those in the control group (S1 Table).

At the end of follow-up in this study, the incidence rate of EE was 0.07% (17/24,360) in patients with DM who were treated with medication, which was significantly lower than the incidence in the control group (0.11%) (P = 0.043). Patients with DM who were treated with medication were older (67.89±13.66 years old) than those in the control group (67.40±13.73 years old), and the difference was statistically significant (P<0.001). Comorbidities of depression, anxiety, and tumors occurred less frequently in patients with DM who were treated with medication than in those in the control group. On the other hand, comorbidities of

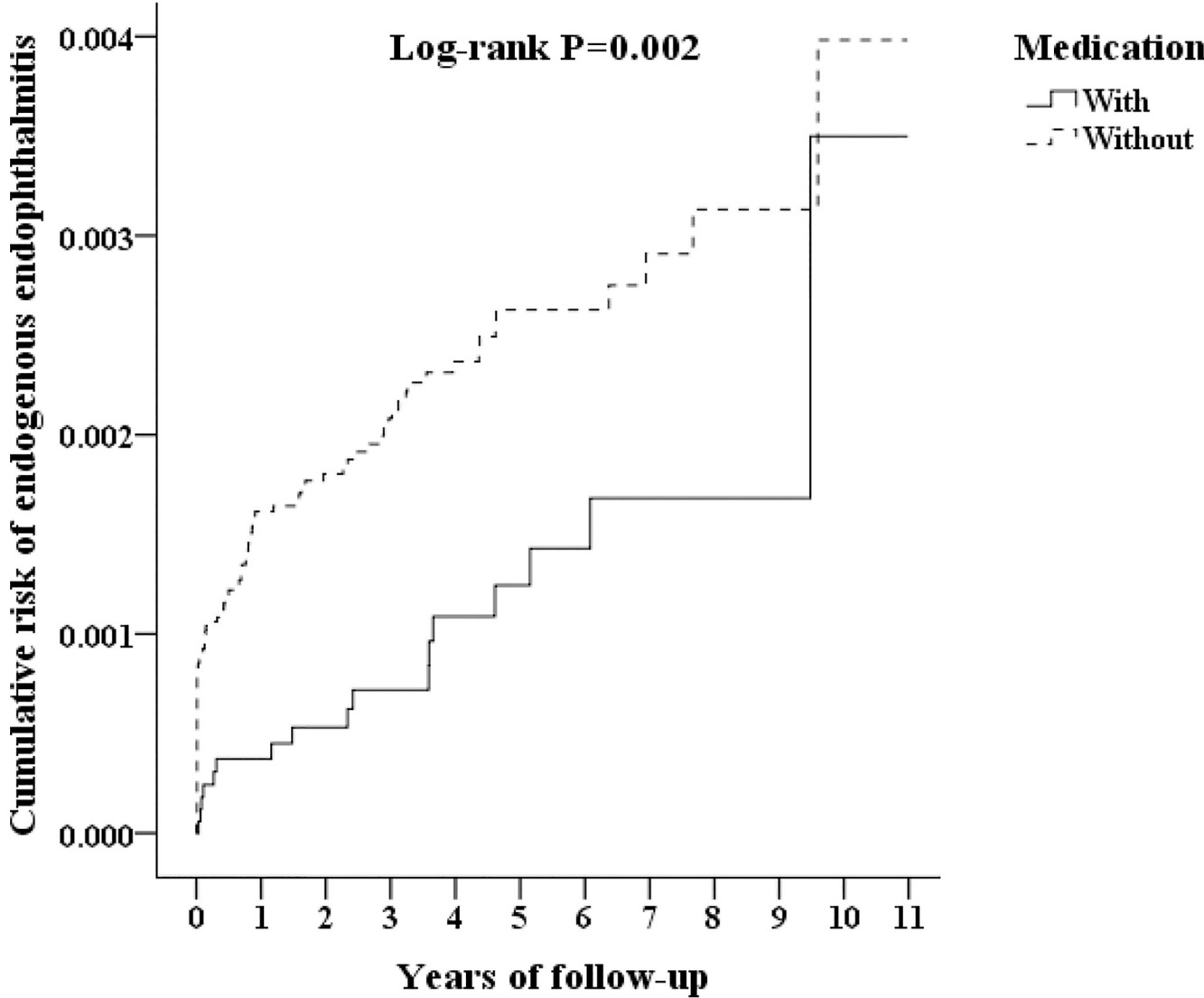

**Fig 2. Kaplan-Meier curve for the cumulative incidence of endogenous endophthalmitis among patients aged 18 and stratified by medication use with the log-rank test.**

hyperlipidemia, septicemia, and pneumonia occurred more frequently in patients with DM who were treated with medication than in those in the control group (S2 Table).

Different factors were analyzed and the adjusted hazard ratios (AHRs) were calculated by using Cox regression (Table 1). Diabetic patients who were treated with medication had lower AHRs (0.47-fold lower) than those in the control group (P = 0.004). Neither sex nor age significantly influenced the AHRs. The group of patients with a comorbidity of liver abscessation had a significant higher AHR (17.615-fold higher) than patients without liver abscesses (P<0.001). Comorbidities of hypertension, renal disease, pneumonia, and tumors, however, had lower AHRs.

Several factors were stratified, and the AHRs were analyzed individually (Table 2) to explore whether specific factors affected the incidence of EE. The results showed that DM patients of both genders who were treated with medication had lower AHRs than those in the control group. Additionally, the female group had a lower AHR than the male group (0.363 vs. 0.562-fold lower). In terms of the age subgroups, all of them showed significantly lower AHRs

**Table 1. Factors of endogenous endophthalmitis determined using cox regression.**

| Variables | Crude HR | 95% CI | 95% CI | P | Adjusted HR | 95% CI | 95% CI | P |
|---|---|---|---|---|---|---|---|---|
| **Medication** | | | | | | | | |
| Without | Reference | | | | Reference | | | |
| With | 0.458 | 0.275 | 0.764 | 0.003 | 0.470 | 0.282 | 0.784 | 0.004 |
| **Sex** | | | | | | | | |
| Male | 1.161 | 0.817 | 1.651 | 0.405 | 1.116 | 0.780 | 1.595 | 0.548 |
| Female | Reference | | | | Reference | | | |
| **Age group (years)** | | | | | | | | |
| 18–44 | Reference | | | | Reference | | | |
| 45–64 | 0.858 | 0.462 | 1.596 | 0.629 | 1.143 | 0.612 | 2.137 | 0.675 |
| ≥65 | 0.444 | 0.237 | 0.831 | 0.011 | 0.747 | 0.392 | 1.422 | 0.375 |
| **Liver abscess** | | | | | | | | |
| Without | Reference | | | | Reference | | | |
| With | 4.534 | 2.441 | 8.419 | <0.001 | 17.615 | 9.236 | 33.595 | <0.001 |
| **HT** | | | | | | | | |
| Without | Reference | | | | Reference | | | |
| With | 0.492 | 0.323 | 0.749 | 0.001 | 0.443 | 0.288 | 0.680 | <0.001 |
| **Depression** | | | | | | | | |
| Without | Reference | | | | Reference | | | |
| With | 1.766 | 0.437 | 7.139 | 0.425 | 1.279 | 0.315 | 5.195 | 0.731 |
| **Anxiety** | | | | | | | | |
| Without | Reference | | | | Reference | | | |
| With | 0.000 | - | - | - | 0.000 | - | - | - |
| **Renal disease** | | | | | | | | |
| Without | Reference | | | | Reference | | | |
| With | 0.282 | 0.104 | 0.764 | 0.013 | 0.273 | 0.100 | 0.742 | 0.011 |
| **Hyperlipidemia** | | | | | | | | |
| Without | Reference | | | | Reference | | | |
| With | 0.000 | - | - | - | 0.000 | - | - | - |
| **Thyrotoxicosis** | | | | | | | | |
| Without | Reference | | | | Reference | | | |
| With | 0.000 | - | - | - | 0.000 | - | - | - |
| **Septicemia** | | | | | | | | |
| Without | Reference | | | | Reference | | | |
| With | 0.727 | 0.392 | 1.350 | 0.313 | 0.713 | 0.376 | 1.352 | 0.300 |
| **Pneumonia** | | | | | | | | |
| Without | Reference | | | | Reference | | | |
| With | 0.379 | 0.177 | 0.813 | 0.013 | 0.373 | 0.171 | 0.874 | 0.013 |
| **Liver disease** | | | | | | | | |
| Without | Reference | | | | Reference | | | |
| With | 0.924 | 0.451 | 1.891 | 0.828 | 1.039 | 0.494 | 2.187 | 0.919 |
| **Tumor** | | | | | | | | |
| Without | Reference | | | | Reference | | | |
| With | 0.214 | 0.079 | 0.580 | 0.002 | 0.164 | 0.060 | 0.446 | <0.001 |
| **CCI_R** | 0.433 | 0.256 | 0.731 | 0.002 | 0.326 | 0.205 | 0.520 | <0.001 |

HR = hazard ratio; CI = confidence interval; Adjusted HR = adjusted hazard ratio (adjusted for the variables listed in the table)

**Table 2. Factors of endogenous endophthalmitis stratified by the variables listed in the table and evaluated using cox regression.**

| Medication (with vs. without) | With | | Without | | Ratio | Adjusted HR | 95% CI | 95% CI | P |
| --- | --- | --- | --- | --- | --- | --- | --- | --- | --- |
| Stratified | Event | Rate (per $10^5$ PYs) | Event | Rate (per $10^5$ PYs) | | | | | |
| **Total** | 17 | 13.172 | 109 | 21.871 | 0.602 | 0.470 | 0.282 | 0.784 | 0.004 |
| **Sex** | | | | | | | | | |
| Male | 11 | 16.632 | 60 | 23.116 | 0.720 | 0.562 | 0.337 | 0.937 | 0.026 |
| Female | 6 | 9.535 | 49 | 20.519 | 0.465 | 0.363 | 0.218 | 0.605 | 0.001 |
| **Age group (years)** | | | | | | | | | |
| 18–44 | 0 | 0.000 | 12 | 43.970 | 0.000 | 0.000 | - | - | - |
| 45–64 | 9 | 21.920 | 51 | 30.435 | 0.720 | 0.562 | 0.337 | 0.938 | 0.029 |
| ≥65 | 8 | 9.724 | 46 | 15.156 | 0.642 | 0.501 | 0.300 | 0.835 | 0.001 |
| **Liver abscess** | | | | | | | | | |
| Without | 16 | 12.631 | 99 | 20.238 | 0.624 | 0.487 | 0.292 | 0.813 | <0.001 |
| With | 1 | 41.759 | 10 | 108.830 | 0.384 | 0.299 | 0.180 | 0.500 | <0.001 |
| **HT** | | | | | | | | | |
| Without | 12 | 15.420 | 86 | 27.961 | 0.551 | 0.430 | 0.258 | 0.718 | <0.001 |
| With | 5 | 9.758 | 23 | 12.054 | 0.809 | 0.632 | 0.379 | 1.054 | 0.296 |
| **Depression** | | | | | | | | | |
| Without | 17 | 13.260 | 107 | 21.678 | 0.612 | 0.477 | 0.286 | 0.796 | <0.001 |
| With | 0 | 0.000 | 2 | 41.785 | 0.000 | 0.000 | - | - | - |
| **Anxiety** | | | | | | | | | |
| Without | 17 | 13.205 | 109 | 21.939 | 0.602 | 0.470 | 0.282 | 0.784 | 0.004 |
| With | 0 | 0.000 | 0 | 0.000 | - | - | - | - | - |
| **Renal disease** | | | | | | | | | |
| Without | 15 | 13.011 | 107 | 24.095 | 0.540 | 0.421 | 0.253 | 0.703 | <0.001 |
| With | 2 | 14.522 | 2 | 3.683 | 3.943 | 3.077 | 0.846 | 9.132 | 0.467 |
| **Hyperlipidemia** | | | | | | | | | |
| Without | 17 | 14.033 | 109 | 23.168 | 0.606 | 0.470 | 0.282 | 0.784 | 0.004 |
| With | 0 | 0.000 | 0 | 0.000 | - | - | - | - | - |
| **Thyrotoxicosis** | | | | | | | | | |
| Without | 17 | 13.402 | 109 | 22.268 | 0.602 | 0.470 | 0.282 | 0.784 | 0.004 |
| With | 0 | 0.000 | 0 | 0.000 | - | - | - | - | - |
| **Septicemia** | | | | | | | | | |
| Without | 15 | 13.116 | 100 | 22.650 | 0.579 | 0.452 | 0.271 | 0.754 | <0.001 |
| With | 2 | 13.603 | 9 | 15.824 | 0.860 | 0.671 | 0.403 | 1.119 | 0.834 |
| **Pneumonia** | | | | | | | | | |
| Without | 15 | 13.475 | 104 | 24.059 | 0.560 | 0.437 | 0.262 | 0.729 | <0.001 |
| With | 2 | 11.272 | 5 | 7.564 | 1.490 | 1.163 | 0.698 | 1.940 | 0.862 |
| **Liver disease** | | | | | | | | | |
| Without | 16 | 13.245 | 102 | 21.906 | 0.605 | 0.472 | 0.283 | 0.787 | <0.001 |
| With | 1 | 12.099 | 7 | 21.383 | 0.566 | 0.442 | 0.265 | 0.737 | <0.001 |
| **Tumor** | | | | | | | | | |
| Without | 17 | 14.823 | 105 | 21.871 | 0.602 | 0.484 | 0.290 | 0.807 | 0.001 |
| With | 0 | 0.000 | 4 | 23.116 | 0.720 | 0.000 | - | - | 0.965 |

PYs = person-years; Adjusted HR = adjusted hazard ratio (adjusted for the variables listed in Table 2); CI = confidence interval.

if treated with medication. Furthermore, all of the comorbidities analyzed in our study displayed no statistically significant differences in AHRs except liver abscess (0.299-fold lower) and liver disease (0.442-fold lower). However, interestingly, patients without comorbidities of

liver abscesses, hypertension, depression, anxiety, renal disease, hyperlipidemia, thyrotoxicosis, septicemia, pneumonia, liver disease, and tumors had lower AHRs among patients with DM who were treated with medication than those in the control group (all P<0.05).

We further stratified several DM medications and analyzed the AHRs separately. The results revealed that patients treated with single-agent DM treatment with insulin (AHR of 0.383, P = 0.001), metformin (AHR of 0.456, P<0.001), gliclazide (AHR of 0.264, P<0.001), glimepiride (AHR of 0.257, P<0.001), or repaglinide (AHR of 0.544, P = 0.019) had lower AHRs than those in the control group. Patients who were treated with combination medical treatment with glimepiride/metformin (AHR of 0.257, P<0.001) or repaglinide/metformin (AHR of 0.544, P = 0.013) also had lower AHRs than those in the control group. However, the rest of the single-agent DM treatments and combinations were not statistically significantly different in terms of AHRs (Table 3).

## Discussion

Endophthalmitis is defined as ocular inflammation particularly affecting the uveal tissue. In most cases in clinical practice, it refers to an intraocular infection caused by microorganisms. According to its pathogenesis, endophthalmitis is divided into endogenous and exogenous subtypes. Several review studies have been published and revealed that EE is associated with many systemic risk factors, including chronic immunocompromising illnesses (DM and renal failure), indwelling or long-term intravenous catheters, immunosuppressive diseases and therapies (malignancies, human immunodeficiency virus infections, and treatment with chemotherapeutic agents), recent invasive surgery, endocarditis, gastrointestinal procedures, hepatobiliary tract infections, and intravenous drug abuse [15–24]. The reason for the occurrence of EE in DM patients may be related to blood-retinal barrier (BRB) breakdown. High serum glucose levels can cause cell dysfunction, and pericytes are the key damaged cells in retinal vessels. Several mechanisms are involved in this diabetic microangiopathy, including loss of tight junction integrity, damage due to advanced glycation end products, oxidative stress, vascular endothelial growth factor synthesis and inflammatory processes. Under these conditions, the BRB becomes disrupted, and vascular permeability is subsequently increased, resulting in substantial leakage or microorganism infiltration through the BRB and into retinal tissue [25–28]. To the best of our knowledge, no studies have evaluated the relationship between glycemic control with different kinds of DM medication and the incidence of EE. In this study, we evaluated 24,360 patients who were newly diagnosed with DM and treated with medications from 2000–2010 in the LHID in Taiwan. The control group was four times larger than the treatment group and included 97,440 patients with DM who were not treated with medication and were matched to the patients in the treatment group (Fig 1).

A KM analysis revealed the cumulative incidence of EE in DM patients who were treated with medication and in those who were not (Fig 2). The incidence of EE was significantly lower risk in the treatment group than in the control group (P = 0.002). Some studies have revealed that an extended duration of DM and elevated serum levels of glycosylated hemoglobin (HbA1c) may impair the protective features of the epidermal barrier and delay wound healing in DM patients [1,29,30]. The correlation between the duration of DM and an increase in endogenous bacterial endophthalmitis was demonstrated via an experimental model, supporting the hypothesis that diabetic ocular changes contribute to the development of endogenous bacterial endophthalmitis [31]. In the present study, we demonstrated that treatment in DM patients can decrease the risk of developing EE; however, this result has not been verified in other studies yet.

Sex and age were matched between the treatment and control groups (P = 0.999) (S1 Table). At the end of follow-up in this study (S2 Table), the proportions of the two sexes

**Table 3. Factors of endogenous endophthalmitis stratified by medication subgroup and evaluated using cox regression.**

| Medication (With vs. without) Medication subgroup | With | | Without | | Ratio | Adjusted HR | 95% CI | 95% CI | P |
|---|---|---|---|---|---|---|---|---|---|
| | Event | Rate (per $10^5$ PYs) | Event | Rate (per $10^5$ PYs) | | | | | |
| Total | 17 | 13.172 | 109 | 21.871 | 0.602 | 0.470 | 0.282 | 0.784 | 0.004 |
| Insulin | 11 | 10.745 | 109 | 21.871 | 0.491 | 0.383 | 0.230 | 0.640 | 0.001 |
| Insulin+lispro | 0 | - | 109 | 21.871 | - | - | - | - | - |
| Insulin+aspart | 4 | 24.097 | 109 | 21.871 | 1.102 | 0.860 | 0.516 | 1.434 | 0.702 |
| Insulin+glulisine | 0 | - | 109 | 21.871 | - | - | - | - | - |
| Insulin+isophane | 0 | - | 109 | 21.871 | - | - | - | - | - |
| Insulin+glargine | 0 | - | 109 | 21.871 | - | - | - | - | - |
| Insulin+detemir | 0 | - | 109 | 21.871 | - | - | - | - | - |
| Metformin | 12 | 12.787 | 109 | 21.871 | 0.585 | 0.456 | 0.274 | 0.761 | <0.001 |
| Gliclazide | 3 | 7.395 | 109 | 21.871 | 0.338 | 0.264 | 0.158 | 0.440 | <0.001 |
| Glimepiride | 3 | 7.199 | 109 | 21.871 | 0.329 | 0.257 | 0.154 | 0.428 | <0.001 |
| Glimepiride+metformin | 3 | 7.199 | 109 | 21.871 | 0.329 | 0.257 | 0.154 | 0.428 | <0.001 |
| Glyburide+metformin | 9 | 17.401 | 109 | 21.871 | 0.796 | 0.621 | 0.373 | 1.036 | 0.384 |
| Sitagliptin+metformin | 1 | 21.865 | 109 | 21.871 | 1.000 | 0.780 | 0.468 | 1.301 | 0.472 |
| Vildagliptin+metformin | 0 | - | 109 | 21.871 | - | - | - | - | - |
| Saxagliptin+metformin | 0 | - | 109 | 21.871 | - | - | - | - | - |
| Linagliptin+metformin | 0 | - | 109 | 21.871 | - | - | - | - | - |
| Repaglinide+metformin | 4 | 15.249 | 109 | 21.871 | 0.697 | 0.544 | 0.326 | 0.908 | 0.013 |
| Acarbose | 0 | - | 109 | 21.871 | - | - | - | - | - |
| Miglitol | 0 | - | 109 | 21.871 | - | - | - | - | - |
| Pioglitazone | 0 | - | 109 | 21.871 | - | - | - | - | - |
| Sitagliptin | 1 | 21.865 | 109 | 21.871 | 1.000 | 0.780 | 0.468 | 1.301 | 0.269 |
| Vildagliptin | 0 | - | 109 | 21.871 | - | - | - | - | - |
| Saxagliptin | 0 | - | 109 | 21.871 | - | - | - | - | - |
| Alogliptin | 0 | - | 109 | 21.871 | - | - | - | - | - |
| Linagliptin | 0 | - | 109 | 21.871 | - | - | - | - | - |
| Exenatide | 0 | - | 109 | 21.871 | - | - | - | - | - |
| Liraglutide | 0 | - | 109 | 21.871 | - | - | - | - | - |
| Dulaglutide | 0 | - | 109 | 21.871 | - | - | - | - | - |
| Dapagliflozin | 0 | - | 109 | 21.871 | - | - | - | - | - |
| Empagliflozin | 0 | - | 109 | 21.871 | - | - | - | - | - |
| Repaglinide | 4 | 15.249 | 109 | 21.871 | 0.697 | 0.544 | 0.326 | 0.908 | 0.019 |
| Nateglinide | 0 | - | 109 | 21.871 | - | - | - | - | - |
| Mitiglinide | 0 | - | 109 | 21.871 | - | - | - | - | - |

PYs = person-years; Adjusted HR = adjusted hazard ratio (adjusted for the variables listed in Table 3); CI = confidence interval.

remained the same as at the baseline, while the distribution of ages did not. Patients in the treatment group were older than those in the control group (P<0.001). In terms of age subgroups, a higher proportion of patients in the treatment group were older (>65 years old) than in the control group (P<0.001). This may have resulted from the protective role of medication and adequate blood sugar control, suggesting that medical treatment might postpone the occurrence of EE in DM patients.

In terms of comorbidities, we analyzed several factors at baseline and at the end of followup. According to DM treatment guidelines, hyperglycemic patients should be treated more aggressively and more strictly when they have comorbidities of hyperlipidemic diseases due to

higher risks of stroke and cardiac events. As a result, hyperlipidemia occurred more frequently in the treatment group than in control group both at baseline (P<0.001) and at the end of follow-up (P<0.001). The proportion of patients with comorbidities of septicemia or pneumonia between groups was not significantly different at baseline. Nevertheless, these two comorbidities were more common in patients in the treatment group at the end of follow-up (P = 0.018 for septicemia and P = 0.028 for pneumonia). This phenomenon might suggest that patients with DM who were being treated with medication had a higher and more poorly controlled glycemic status than those who were not being treated with medication; thus, these patients were more likely to suffer from infection (S1 and S2 Tables).

We further analyzed the AHRs of several factors, as shown in Table 1. Among the factors we included in this study, medical treatment was the most important one. To date, some studies have suggested that DM is one of the most important predisposing factors for developing endophthalmitis and that it is a worse prognostic indicator [8,32,33]. DM compromises the immune system, increases the risk of infection, causes architectural changes in the eye, and enhances the ability of organisms to invade the eye [1,31,34]. Therefore, in clinical practice, we suggest that blood sugar in DM patients should be aggressively controlled to diminish the incidence of further complications. However, no studies have yet demonstrated whether treatment with medication or not in DM patients affects the incidence of EE. In the present study, we found that the AHR was 0.47-fold lower in the treatment group than in the control group (P = 0.004). This result suggests that doctors in clinical practice should treat DM patients earlier or more strictly to prevent EE. Neither sex nor age was significantly different in terms of AHRs. Nevertheless, some factors were still significantly different. First, a history of liver abscesses was the only factor that increased AHR significantly in our study. This is understandable because in Asian countries a high proportion of endogenous endophthalmitis is caused by Klebsiella pneumonia liver abscesses [17,35]. Accordingly, patients diagnosed with liver abscesses tended to have a higher risk of developing EE. Second, patients with comorbidities of hypertension, renal disease, or neoplastic disease had lower AHRs than those without these comorbidities (0.443-fold, 0.273-fold, 0.164-fold lower, respectively). This may be because these patients need regular clinical evaluations. At each clinical evaluation, patients may mention discomfort or symptoms, particularly new-onset disturbances. These complaints may alert clinicians to consider more diagnostics. Third, a comorbidity of pneumonia was associated with a lower AHR (0.373-fold lower, P = 0.013) in terms of the incidence of EE. Most cases of pneumonia were caused by bacterial infections and were treated with antibiotics. In patients with more severe pneumonia, those with more comorbidities, or those treated at tertiary medical centers, a broad-spectrum antibiotic was usually prescribed. This phenomenon is particularly prominent in Taiwan. Therefore, antibiotics in the bloodstream in pneumonia patients may play a therapeutic role during infectious episodes, such as during EE episodes (Table 1).

We further stratified several variable factors and analyzed their AHRs between the treatment and control groups. Some interesting results were found. First, in DM patients of both genders, those treated with medication had lower AHRs (0.562-fold in males and 0.363-fold in females) than DM patients who were not treated with medication. Second, in DM patients with the comorbidities of liver abscess and liver disease, the treatment group had lower AHRs (0.299 and 0.442-fold) than the control group. Thus, we can emphasize the importance of medication in DM patients especially in those with comorbid liver abscess and liver disease. Third, for rest of the diseases we analyzed, patients without these comorbidities (such as hypertension, renal disease, septicemia, pneumonia, and tumors) had lower AHRs in terms of the incidence of EE if they were treated with DM medications (all P<0.05). DM patients with other underlying diseases usually had worse in systemic conditions and bore more disease burdens. On the other hand, uncomplicated DM individuals had more advantages in terms of

decreased AHRs for EE if they were treated with DM medications, according to this finding. Therefore, DM medication may be a protective factor for EE particularly in DM patients without comorbidities (Table 2).

Different DM medications were also utilized to stratify the patients and analyzed (Table 3). We found that single-agent therapy with some specific medications, such as insulin, metformin, gliclazide, glimepiride, and repaglinide, were associated with lower AHRs. To explain this finding, we may assume that single-agent treatment is usually applied for DM patients during the early stages of disease. Patients with early-stage DM can be controlled well and can have a lower risk of developing EE. On the other hand, combination therapy with glimepiride/metformin and repaglinide/metformin was also associated with lower AHRs (0.257 and 0.544-fold, respectively). This result suggests that clinicians should choose the above medications, no matter single or combination therapy, in patients with advanced DM for the purpose of diminishing the incidence of EE. Although the number of events were relatively low in the present study due to the low incidence of EE, these results may still be valuable as a reference in clinical practice.

There are some limitations of the present study. First, this study was conducted with a cohort, case-comparative method, which has less statistical reliability than randomized controlled trials. Second, NHIRD does not offer detailed clinical information regarding clinical manifestations or disease severity, and laboratory findings are lacking, especially HbA1c data, which are particularly essential in diagnosing DM and monitoring disease progression. We wish to conduct another retrospective study in the future to investigate the relationship between HbA1c and EE incidence.

## Conclusions

We searched a nationwide research database and evaluated 24,360 DM patients who were treated with medication and a 4 times larger comparison group of patients who were not treated with medication over a span of 11 years. An incidence rate of EE of 0.07% was found in the treatment group. Further, patients in the treatment group had a lower risk of developing EE than those in the control group (log-rank test, P = 0.002). DM patients who were treated with medication had lower AHRs (0.47-fold lower) than those in the control group (P = 0.004). When several factors were stratified individually, we discovered that DM patients with comorbid liver abscess or liver disease who were treated with medication had lower AHRs than those in the control group. In terms of the DM medication stratification test, single-agent therapy with insulin, metformin, gliclazide, glimepiride, and repaglinide and combination therapy with glimepiride/metformin and repaglinide/metformin had lower AHRs.

## Supporting information

**S1 Table. Characteristics of the study sample at baseline.**
(DOC)

**S2 Table. Characteristics of the study sample at the end of follow-up.**
(DOC)

## Author Contributions

**Conceptualization:** Ke-Hung Chien, Ke-Hao Huang, Yun-Hsiu Hsieh, Chang-Min Liang, Tzu-Heng Weng, Wu-Chien Chien.

**Data curation:** Chi-Hsiang Chung, Wu-Chien Chien.

**Formal analysis:** Chi-Hsiang Chung, Wu-Chien Chien.

**Investigation:** Ke-Hung Chien, Chi-Hsiang Chung, Tzu-Heng Weng, Wu-Chien Chien.

**Methodology:** Chi-Hsiang Chung, Tzu-Heng Weng, Wu-Chien Chien.

**Software:** Chi-Hsiang Chung.

**Supervision:** Ke-Hung Chien, Chang-Min Liang, Wu-Chien Chien.

**Validation:** Chi-Hsiang Chung.

**Writing – original draft:** Ke-Hung Chien, Yu-Hua Chang, Tzu-Heng Weng.

**Writing – review & editing:** Ke-Hung Chien, Chang-Min Liang, Yu-Hua Chang, Tzu-Heng Weng, Wu-Chien Chien.

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
