## [Decision Letter · Decision Letter 0]

4 Nov 2019

PONE-D-19-23998

The impact of diabetes mellitus medication on the incidence of endogenous endophthalmitis

PLOS ONE

Dear Dr. Weng,

Thank you for submitting your manuscript to PLOS ONE. After careful consideration, we feel that it has merit but does not fully meet PLOS ONE’s publication criteria as it currently stands. Therefore, we invite you to submit a revised version of the manuscript that addresses the points raised during the review process.

The pateint popultion needs to be better described, and more describtion of the mechanim is needed. 

We would appreciate receiving your revised manuscript by Dec 19 2019 11:59PM. To enhance the reproducibility of your results, we recommend that if applicable you deposit your laboratory protocols in protocols.io, where a protocol can be assigned its own identifier (DOI) such that it can be cited independently in the future. For instructions see: http://journals.plos.org/plosone/s/submission-guidelines#loc-laboratory-protocols

We look forward to receiving your revised manuscript.

Kind regards,

Andrew W Taylor, Ph.D.

Academic Editor

PLOS ONE

Journal Requirements:

This study was supported by grants from the Tri-Service General Hospital Research Foundation (TSGH-C107-004 and TSGH-C107-090) and the Ministry of National Defense Medical Affairs Bureau (MAB-108-049). The sponsors had no roles in the study design, data collection and analysis, decision to publish, or preparation of the manuscript.

Reviewers' comments:

Reviewer's Responses to Questions

**Comments to the Author**

1. Is the manuscript technically sound, and do the data support the conclusions?

Reviewer #1: Partly

2. Has the statistical analysis been performed appropriately and rigorously? 

Reviewer #1: Yes

3. Have the authors made all data underlying the findings in their manuscript fully available?

Reviewer #1: Yes

4. Is the manuscript presented in an intelligible fashion and written in standard English?

Reviewer #1: Yes

5. Review Comments to the Author

Reviewer #1: In this manuscript, Chien et al. describe the results of a matched cohort study of the incidence of endogenous endophthalmitis (EE) among patients diagnosed with diabetes mellitus (DM). The objective was to determine whether treatment of diabetes is associated with a lower incidence of EE. The authors evaluated 121,800 patients with DM. Among these patients, 24,360 were treated with medication, and 97,440 were not. Single-agent therapy with insulin, metformin, gliclazide, glimepiride, or repaglinide and combination therapy with glimepiride/metformin or repaglinide/metformin were associated with a decreased risk of EE. Neither sex nor age influenced risk, however those patients with liver abscesses were at increased risk relative to patients without liver abscesses. There are a number of issues with this manuscript. The first is that no information was provided on patient diagnosis such as A1C levels or whether the patients were type 1 or 2. Second, it also is not clear why so many patients with diabetes were not being treated. Were they all under dietary control for diabetes, or were they without medication due to lack of compliance or lack of medical care? Finally, the authors were not clear on their rationale for performing this study. Indeed, diabetes mellitus is an underlying risk factor for developing endogenous endophthalmitis, however they were vague in describing the mechanism for this. Another group has demonstrated that this correlation is likely due to the break down of the blood retinal barrier during diabetes development and progression to diabetic rertinopathy, and likely has little to do with keratitis.

6. PLOS authors have the option to publish the peer review history of their article (what does this mean?). If published, this will include your full peer review and any attached files.

Reviewer #1: No

---

## [Author Response · Author response to Decision Letter 0]

19 Nov 2019

Dear Editor-in-Chief, 

We very much appreciate the comments and critiques from the reviewers regarding Manuscript ID PONE-D-19-23998, entitled "The impact of diabetes mellitus medication on the incidence of endogenous endophthalmitis". We have performed the additional work requested by the reviewers to improve our manuscript. We have also written point-by-point responses to the reviewers’ comments. Overall, the manuscript has been substantially revised to reflect each of the reviewers’ suggestions. All coauthors have read and approved this manuscript and agreed to fully transfer its copyright to PLOS ONE. I sign for and accept responsibility for this material on behalf of all coauthors. We hope that these changes will make the paper more impactful and that it can be published in your journal in the near future.

Sincerely,

Ke-Hung Chien, MD

Tzu-Heng Weng, MD 

Department of Ophthalmology, Tri-Service General Hospital, National Defense Medical Center, Taipei, Taiwan; No. 325, Section 2, Cheng-Kung Road, Neihu District, Taipei City, 11490, Taiwan, Republic of China.

---

## [Decision Letter · Decision Letter 1]

19 Dec 2019

The impact of diabetes mellitus medication on the incidence of endogenous endophthalmitis

PONE-D-19-23998R1

Dear Dr. Weng,

We are pleased to inform you that your manuscript has been judged scientifically suitable for publication and will be formally accepted for publication once it complies with all outstanding technical requirements.

With kind regards,

Andrew W Taylor, Ph.D.

Academic Editor

PLOS ONE

Additional Editor Comments (optional):

Reviewers' comments:

Reviewer's Responses to Questions

**Comments to the Author**

1. If the authors have adequately addressed your comments raised in a previous round of review and you feel that this manuscript is now acceptable for publication, you may indicate that here to bypass the “Comments to the Author” section, enter your conflict of interest statement in the “Confidential to Editor” section, and submit your "Accept" recommendation.

Reviewer #1: All comments have been addressed

2. Is the manuscript technically sound, and do the data support the conclusions?

Reviewer #1: (No Response)

3. Has the statistical analysis been performed appropriately and rigorously? 

Reviewer #1: (No Response)

4. Have the authors made all data underlying the findings in their manuscript fully available?

Reviewer #1: (No Response)

5. Is the manuscript presented in an intelligible fashion and written in standard English?

Reviewer #1: (No Response)

6. Review Comments to the Author

Reviewer #1: (No Response)

7. PLOS authors have the option to publish the peer review history of their article (what does this mean?). If published, this will include your full peer review and any attached files.

Reviewer #1: No

---

## [Editor Report · Acceptance letter]

26 Dec 2019

PONE-D-19-23998R1 

The impact of diabetes mellitus medication on the incidence of endogenous endophthalmitis 

Dear Dr. Weng:

I am pleased to inform you that your manuscript has been deemed suitable for publication in PLOS ONE. Congratulations! Your manuscript is now with our production department. 

With kind regards,

on behalf of

Dr. Andrew W Taylor 

Academic Editor

PLOS ONE